# Effects of Voluntary Wheel Running Exercise on Chemotherapy-Impaired Cognitive and Motor Performance in Mice

**DOI:** 10.3390/ijerph20075371

**Published:** 2023-04-03

**Authors:** Thomas H. Lee, Malegaddi Devaki, Douglas A. Formolo, Julia M. Rosa, Andy S. K. Cheng, Suk-Yu Yau

**Affiliations:** 1Department of Rehabilitation Sciences, Hong Kong Polytechnic University, Hung Hom, Hong Kong; hyt.lee@outlook.com (T.H.L.); devaki.chm@gmail.com (M.D.); douglas.formolo@connect.polyu.hk (D.A.F.); julia.macedorosa@polyu.edu.hk (J.M.R.); 2Mental Health Research Center (MHRC), Hong Kong Polytechnic University, Hung Hom, Hong Kong; 3Research Institute for Smart Aging (RISA), Hong Kong Polytechnic University, Hung Hom, Hong Kong

**Keywords:** physical exercise, chemotherapy, adult hippocampal neurogenesis, depression, muscle wasting, learning and memory

## Abstract

Chemotherapy-induced cognitive impairment (chemobrain) and muscle wasting (cachexia) are persisting side effects which adversely affect the quality of life of cancer survivors. We therefore investigated the efficacy of physical exercise as a non-pharmacological intervention to reverse the adverse effects of chemotherapy. We examined whether physical exercise in terms of voluntary wheel running could prevent chemotherapy-induced cognitive and motor impairments in mice treated with the multi-kinase inhibitor sorafenib. Adult male BALB/c mice were subdivided into runner and non-runner groups and orally administered with sorafenib (60 mg/kg) or vehicle continuously for four weeks. Mice could freely access the running wheel anytime during sorafenib or vehicle treatment. We found that sorafenib treatment reduced body weight gain (% of change, vehicle: 3.28 ± 3.29, sorafenib: −9.24 ± 1.52, *p* = 0.0004), impaired hippocampal-dependent spatial memory in the Y maze (exploration index, vehicle: 35.57 ± 11.38%, sorafenib: −29.62 ± 7.90%, *p* < 0.0001), increased anhedonia-like behaviour in the sucrose preference test (sucrose preference, vehicle: 66.57 ± 3.52%, sorafenib: 44.54 ± 4.25%, *p* = 0.0005) and impaired motor skill acquisition in rotarod test (latency to fall on day 1: 37.87 ± 8.05 and day 2: 37.22 ± 12.26 s, *p* > 0.05) but did not induce muscle wasting or reduce grip strength. Concomitant voluntary running reduced anhedonia-like behaviour (sucrose preference, sedentary: 44.54 ± 4.25%, runners: 59.33 ± 4.02%, *p* = 0.0357), restored impairment in motor skill acquisition (latency to fall on day 1: 50.85 ± 15.45 and day 2: 168.50 ± 37.08 s, *p* = 0.0004), but failed to rescue spatial memory deficit. Immunostaining results revealed that sorafenib treatment did not affect the number of proliferating cells and immature neurons in the hippocampal dentate gyrus (DG), whereas running significantly increased cell proliferation in both vehicle- (total Ki-67^+^ cells, sedentary: 16,687.34 ± 72.63, exercise: 3320.03 ± 182.57, *p* < 0.0001) and sorafenib-treated mice (Ki-67^+^ cells in the ventral DG, sedentary: 688.82.34 ± 38.16, exercise: 979.53 ± 73.88, *p* < 0.0400). Our results suggest that spatial memory impairment and anhedonia-like behaviour precede the presence of muscle wasting, and these behavioural deficits are independent of the changes in adult hippocampal neurogenesis. Running effectively prevents body weight loss, improves motor skill acquisition and reduces anhedonia-like behaviour associated with increased proliferating cells and immature neurons in DG. Taken together, they support physical exercise rehabilitation as an effective strategy to prevent chemotherapy side effects in terms of mood dysregulation and motor deficit.

## 1. Introduction

Chemotherapy-induced cognitive impairment (chemobrain), muscle wasting, and weakness (cachexia) are commonly diagnosed in cancer patients [1,2,3,4]. Impaired short-term memory, working memory, verbal ability, visuospatial memory, executive functions, and attention span are observed in clinical subjects undergoing chemotherapy [5,6,7]. On the other hand, involuntary weight loss, specifically muscle mass loss [8,9], in the early stage of chemotherapy correlates with lower survival outcomes in cancer patients [10,11,12,13,14]. Post-therapy quality of life can be severely affected by persisting cognitive impairments [5] and myopathy [15,16] after chemotherapy discontinuation, while other psychological distress, like depressive disorder, may also be arisen [17]. Importantly, rodent studies have shown that chemotherapy alone can contribute to cognitive deficits [18] and muscle wasting [19,20] in non-tumour-bearing animals, implicating the potential neurotoxicity and myotoxicity brought by chemotherapy to cancer patients.

Sorafenib is a common chemo drug, acting as a multi-kinase inhibitor for patients diagnosed with advanced clear-cell renal and hepatocyte carcinoma [21,22]. Sorafenib suppresses tumour cell proliferation and survival through RAF/MEK/ERK cascade [23]. It also suppresses angiogenesis by inhibiting vascular endothelial growth factor (VEGF) and platelet-derived growth factor (PDGF) receptors [24]. Although sorafenib promotes patient survivorship [25], its apoptotic [26] and oxidative properties [27] may contribute to cognitive impairment [28,29] and muscular dysfunction [30,31]. Sorafenib can cross the blood-brain barrier [32], through which sorafenib can directly impair brain function [26,33].

The hippocampus is a neurogenic zone with active adult neurogenesis in the subregion dentate gyrus (DG), playing an important role in regulating learning and memory, as well as mood [34,35]. Previous studies have demonstrated that sorafenib administration induces motor and cognitive deficits and impairs hippocampal function [26,33]. In contrast, physical exercise is effective in preventing cancer cachexia and cognitive impairments [36], improving cancer survivorship and chemotherapy-induced muscle wasting [37,38,39,40,41]. As shown in rodent models, exercise also improves hippocampal impairment and cognitive deficits [42,43,44]. Muscle loss and muscle weakness are often associated with cognitive deficits [45,46,47].

In this study, we aim to investigate whether chemotherapy-impaired motor performance, mood regulation, learning and memory can be reversed by physical wheel running in a mouse model. We further examine whether behavioural changes are associated with changes in adult hippocampal neurogenesis. We hypothesise that voluntary running exercise prevents chemotherapy-induced muscle wasting and improves learning and memory impairments associated with changes in hippocampal neurogenesis. We, therefore, hypothesise that: (1) concomitant voluntary running exercise not only prevents chemotherapy-induced muscle wasting but also improves learning and memory impairments; and (2) changes of hippocampal neurogenesis are involved in behavioural improvement following an exercise intervention in mice that received chemotherapy.

## 2. Materials and Methods

### 2.1. Animals and Experimental Design

Male BALB/c mice were provided by the Centralised Animal Facility at The Hong Kong Polytechnic University. Mice were maintained under a 12 h light-dark cycle with full access to food and water. Eight-week-old mice were randomly assigned to four experimental groups: Vehicle + Non-runners, Vehicle + Runners, Sorafenib + Non-runners, and Sorafenib + Runners. After treatment, the first cohort of mice underwent behavioural assessments of learning and memory performance and affective behaviours (Figure 1A), while the second cohort underwent skeletal muscle strength examination (Figure 1B). Based on our pilot experiment investigating the impact of sorafenib treatment on cognition (Cohen’s d = 1.60), we estimated that eight animals per group would be necessary to achieve a study power of 80%, considering a level of significance to be 5%. All experimental procedures were approved and followed the guidelines of the Animal Subjects Ethics Sub-Committee at The Hong Kong Polytechnic University.

### 2.2. Drug Preparation and Treatment

Sorafenib (APE×Bio, Houston, TX, USA) was first dissolved in DMSO at 200 mg/mL and then resuspended in 0.5% carboxymethylcellulose salt and 0.25% Tween-80 immediately before oral gavage [48]. Sorafenib (60 mg/kg, p.o.), or vehicle, was administered continuously for 28 days.

### 2.3. Voluntary Wheel Running Exercise

Voluntary wheel running was carried out as previously described [49]. Mice were pair-housed in a cage to avoid additional stress induced by social isolation. Two runner mice shared a cage equipped with a running wheel and could freely access the wheel 24 h per day over 28 days of treatment. Daily running wheel activity was recorded as the number of revolutions per day over the 28-d running period. The activity wheel dictator (CKC TINNER, Beijing, China) converted the number of revolutions (34.55 cm inner circumference) to the total running distance. Wheels were locked during behavioural assessments in the runners’ cages.

### 2.4. Behavioural Assessments

Mice were briefly handled by the experimenter over the drug treatment period. On the testing days, mice were allowed to habituate in the behavioural testing room for 2 h before experiments.

#### 2.4.1. Y-Maze Task (YMT)

To examine the hippocampal-dependent spatial memory, mice were subjected to the three-arm Y-maze test (arm dimensions: L × W × H: 10 × 6 × 8 cm) as previously reported [50]. The start arm (S), familiar arm (F), and novel arm (N) were counterbalanced to avoid spatial recognition bias. The test was divided into two phases. In phase 1, a mouse could explore the maze for 10 min with the novel arm blocked. After a 3 h intermission, the mouse returned to phase 2, where the novel arm was unblocked and could be freely explored for 5 min. The exploration index was calculated as (time spent in N—time spent in F)/(total time spent in N + F); and presented on a −1 to 1 scale.

#### 2.4.2. Sucrose Preference Test (SPT)

Rodents have an innate preference for sweet-tasting food and drinks [51], whereas stressors can induce anhedonia-like behaviour, as indicated by a reduced tendency to seek palatable rewards [52]. To examine anhedonia-like behaviour, mice were individually housed for 24 h with access to food pellets and two water bottles, one containing tap water and the other containing 1% sucrose solution, as a palatable reward. The positions of the bottles were swapped after 12 h to avoid side bias. Bottles were pre-weighed before the experiment and re-weighed 24 h after to calculate the net consumption of sucrose solution and water. Sucrose preference was measured in percentage as sucrose solution consumption/total fluid consumption [49].

#### 2.4.3. Open Field Test (OFT)

To examine locomotor function, mice were allowed to freely explore a square open-field arena (L × W × H: 40 × 40 × 30 cm^3^) for 15 min, as previously described [50]. Locomotor activity in the open field box was video-taped. Distance travelled and mean speed over the first 5 min and the total 15 min were automatically detected and calculated by the ANYMAZE software (Stoelting Co., Wood Dale, IL, USA). Between tests, the arena was cleaned with 70% ethanol to eliminate olfactory cues.

#### 2.4.4. Forced Swim Test (FST)

The test examines coping strategies adopted by the mice under acute inescapable stress [53]. Struggling and swimming are active coping strategies to avoid adverse outcomes, such as drowning, whereas immobility is a passive coping strategy indicative of depression-like behaviour, as antidepressants tend to increase activity to the detriment of passive coping strategies. Mice were placed in a transparent cylinder (height: 30 cm; radius: 15 cm) filled with 2/3 water (23–25 °C) for 6 min [49]. The test was video-taped, and a trained experimenter manually scored the time spent immobile during the last 4 min of the test in an observer-blinded manner. Immobility was considered as the absence of any movement but those necessary for keeping the head above the water [49].

#### 2.4.5. Rotarod Test (RR)

Muscle strength, coordination, and balance were assessed using a rotarod apparatus (Panlab, Barcelona, Spain) [54]. Mice were trained for two consecutive days with three trials per day before the test. Each trial lasted a maximum of 5 min and was separated by a 15 min intermission. During training, mice maintained their balance on a stationary spindle for 30 s, followed by habituating on the rotating spindle at a constant speed of 4 rpm. On the test day, mice returned to the rotarod at an initial speed of 8 rpm followed by 0.3 rpm/s acceleration until fall. The trial was repeated after a 15 min intermission if the mouse fell from the rotarod within 3 s. The average latency to fall was recorded.

### 2.5. Grip Strength Test (GST)

To measure grip strength, mice were required to grasp with all four limbs or the forelimbs only. A horizontal grid connected to a dynamometer (KEW, Beijing, China) while pulled backwards five times [55]. The force applied to the grid was recorded in newton. The mean of the five measurements was normalised by their body weight.

### 2.6. Inverted Grid Hanging Test (IGT)

Functional assessment of muscle strength was further examined by placing the mouse at the centre of an invertible wire grid. The grid was elevated 50–60 cm above a soft underlay to prevent jumping. The grid was gently shaken three times to ensure the mouse firmly grasped the wires. Then, the grid was inverted, and the latency to falls was recorded. Mice were tested three times with a 30 min intermission [56].

### 2.7. Immunohistochemistry

The day after behavioural assessments, the mice were deeply anaesthetised with isoflurane and then perfused with phosphate-buffered saline (0.01 M PBS; pH 7.4) followed by 4% paraformaldehyde (PFA). The brains were extracted and stored in 4% PFA overnight and transferred into 30% sucrose until they sank. Coronal sections (1-in-6 series, 30-μm thick) were prepared using a vibratome (Leica Biosystems, Nußloch, Germany). The brain slices were preserved in cryoprotectant solution (30% glycerol and 30% ethylene glycol) at 4 °C. Antigens were retrieved by citric acid buffer (pH 6.0) at 95 °C for 10 min. The sections were incubated with primary antibody anti-Ki-67 (1:1000; abcam, Cambridge, UK) or anti-DCX (1:200; Santa Cruz Biotechnology, Dallas, TX, USA) overnight at room temperature, followed by biotinylated goat anti-rabbit IgG or goat anti-mouse IgG (1:200; Vector Laboratories, Burlingame, CA, USA) for 2 h. Finally, the sections were visualised using the peroxidase method with commercially available DAB kits (Vector Laboratories, Burlingame, CA, USA).

### 2.8. Quantifications of Ki-67 and DCX-Immunopositive Cells

Total number of DCX- or Ki-67-immunopositive cells present in the subgranular zone (SGZ) of the dorsal DG (bregma −1.34 to −2.54) and the ventral DG (bregma −2.54 to −3.40) were quantified in a sample-blinded manner using a bright-field microscope (Nikon, Tokyo, Japan) by multiplying the number of immunopositive cells per DG section by the number of brain sections containing in the dorsal or the ventral DG, as previously performed and described [57].

### 2.9. Isolation of Skeletal Muscle Tissues

The tibialis anterior (TA), extensor digitorum longus (EDL), gastrocnemius (GA) and soleus (SOL) muscles from the hind limbs were dissected as previously described [58]. The hind limb muscles were fully exposed by cutting the skin and the fascia of the legs. To expose the EDL muscle, the distal TA tendon at the ankle was cut to peel the TA muscle upwards towards the proximal tendon. Similarly, the EDL muscle was removed from the distal to the proximal end. The hindlimb was flipped to the posterior side to expose the GA muscle. The Achilles tendon was cut from the ankles to remove GA. The thin, dark-red SOL muscle along the centre of the anterior surface of GA was exposed. The wet masses of skeletal muscles were immediately weighed.

### 2.10. Statistical Analyses

Two-way analyses of variance (ANOVA) with Tukey’s post-hoc test were used to compare the effects of sorafenib and voluntary wheel running exercise. Datasets with *p* ≥ 0.05 in the Shapiro-Wilk test for ANOVA residuals were considered not to be deviated from normality. An unpaired *t*-test with Welch’s correction was used to compare the average daily running distance between the Vehicle + Runners group and Sorafenib + Runners group after testing the normality of the two groups with Shapiro–Wilk test. Three-way ANOVA with Tukey’s post-hoc test was used to evaluate motor learning by comparing the interaction among the effects of sorafenib, voluntary running exercise, and days of training in the rotarod training. All data were analysed by GraphPad Prism 9 software (GraphPad Software, San Diego, CA, USA). All graphs were shown as mean ± SEM. *p* < 0.05 was considered statistically significant with a 95% confidence interval.

## 3. Results

### 3.1. Voluntary Running Exercise Attenuated Sorafenib-Induced Weight Loss

The ANOVA residual of body weight change was normally distributed (Figure 1C; W = 0.9561; *p* = 0.1419). Two-way ANOVA revealed a significant effect of sorafenib on body weight change (Figure 1C; F_1,34_ = 61.46, *p* < 0.0001), although there was no interaction between sorafenib administration and voluntary running (Figure 1C; F_1,34_ = 2.48, *p* = 0.1615). Sorafenib treatment induced significant body weight loss in non-runners compared with vehicle-treated non-runners (Figure 1C; Tukey’s post-hoc: *p* = 0.0004 vs. Vehicle + Non-runners). Conversely, concomitant voluntary wheel running showed the main effect on body weight change (Figure 1C; F_1,34_ = 59.42, *p* < 0.001), by which it increased body weight gain in vehicle-treated runners (Figure 1C; Tukey’s post-hoc: *p* < 0.0001 vs. Vehicle + Non-runners). Although the average daily running activities of sorafenib-treated runners were lower than those of vehicle-treated runners (Figure 1D; t_41.59_ = 2.620, *p* = 0.0122), voluntary running attenuated sorafenib-induced body weight loss (Figure 1C; Tukey’s post-hoc: *p* = 0.0007 vs. Sorafenib + Non-runners).

### 3.2. Sorafenib Administration Impaired Spatial Memory

The ANOVA residual of the exploration index in the Y-maze test was normally distributed (Figure 2A; W = 0.95; *p* = 0.1757). Sorafenib had significant main effects on spatial learning and memory performance (Figure 2A; F_1,26_ = 49, *p* < 0.0001), while there was no interaction between sorafenib administration and voluntary running (Figure 2A; F_1,26_ = 1.6, *p* = 0.2134). Sorafenib administration impaired spatial working memory in non-runners (Figure 2A; Tukey’s post-hoc: *p* < 0.0001 vs. Vehicle + Non-runners). Voluntary running exerted a significant effect on spatial memory preference in sorafenib-treated runners (Figure 2A; F_1,26_ = 8.1, *p* = 0.0086; Tukey’s post-hoc: *p* = 0.0345 vs. Sorafenib + Non-runners), but spatial memory impairments were still present in sorafenib-treated runners since they did not show any preference towards familiar and novel arms when compared to the novelty preference of vehicle-treated non-runners (Figure 2A; Tukey’s post-hoc: *p* = 0.0383 vs. Vehicle + Non-runners).

Anhedonia-like and depression-like behaviours were assessed by SPT and FST, respectively. The ANOVA residuals in SPT (Figure 2B; W = 0.9580; *p* = 0.1866) fell into normality. In SPT, sorafenib administration exerted a significant effect on sucrose preference (Figure 2B; F_1,32_ = 20.27, *p* < 0.0001). Sorafenib treatment increased anhedonia-like behaviour as indicated by reduced sucrose preference in non-runner when compared to vehicle-treated counterparts (Figure 2B; Tukey’s post-hoc: *p* = 0.0005 vs. Vehicle + Non-runners), whereas voluntary running reduced anhedonia-like behaviour (Figure 2B; Tukey’s post-hoc: *p* = 0.0357 vs. Sorafenib + Non-runners; effect of exercise: F_1,32_ = 6.06, *p* = 0.0194).

In FST, running exercise had a strong effect on reducing depressive behaviours (Figure 2C; F_1,31_ = 47.05, *p* < 0.0001), but not sorafenib treatment (F_1,31_ = 1.69, *p* = 0.2024), and there was no interaction effect (Figure 2C; F_1,31_ = 2,639, *p* = 0.2031). Voluntary running significantly reduced immobility time in both vehicle-treated (Figure 2C; Tukey’s post-hoc: *p* < 0.0001 vs. Vehicle + Non-runners) and sorafenib-treated mice (Figure 2C; Tukey’s post-hoc: *p* = 0.0363 vs. Sorafenib + Non-runners) when compared to non-runner counterparts. Although the ANOVA residuals in the FST suggested non-normal distribution, a post hoc analysis using the Mann-Whitney comparison test confirmed significant effects of running for vehicle- (Mann-Whitney U = 3, *p* = 0.0002 vs. Non-runners) and sorafenib-treated (Mann-Whitney U = 12, *p* = 0.0379 vs. Non-runners) mice.

### 3.3. Voluntary Running Restored Motor Learning Deficit Induced by Sorafenib

The ANOVA residuals in the latency to fall were normally distributed (Figure 3B; W = 0.9839; *p* = 0.6359). Three-way ANOVA revealed significant the main effects for sorafenib (Figure 3A; F_1,25_ = 26.37, *p* < 0.0001), voluntary running (Figure 3A; F_1,25_ = 17.16, *p* = 0.0003), and days of rotarod training (Figure 3A; F_1,25_ = 72.53, *p* < 0.0001). Additionally, there were significant interactions among the three factors (Figure 3A; F_1,25_ = 4.48, *p* = 0.0444). In terms of motor learning, both vehicle-treated non-runners (Figure 3A; Tukey’s post-hoc: *p* = 0.0025 vs. Day 1: Vehicle + Non-runners) and runners (Figure 3A; Tukey’s post-hoc: *p* < 0.0001 vs. Day 1: Vehicle + Runners) had higher latency to fall on the second day of rotarod training. In particular, running significantly delayed the latency to fall on the second day in vehicle-treated mice (Figure 3A; Tukey’s post-hoc: *p* = 0.0037 vs. Day 2: Vehicle + Non-runners). Sorafenib treatment did not show significant effect on latency for any mice on the second day (Figure 3A; Tukey’s post-hoc: *p* > 0.05 vs. Day 1: Sorafenib + Non-runners), but vehicle-treated non-runners showed a delay in latency to fall (Figure 3A; Tukey’s post-hoc: *p* < 0.0001 vs. Day 2: Vehicle + Non-runners). Conversely, voluntary wheel running increased latency to fall in sorafenib-treated mice (Figure 3A; Tukey’s post-hoc: *p* = 0.0004 vs. Day 1: Sorafenib + Runners; *p* = 0.0004 vs. Day 2: Sorafenib + Non-runners), suggesting an improvement in motor learning deficits.

We further examined the motor coordination with an accelerated rotarod on the third day. The ANOVA residuals in the latency to fall were normally distributed (Figure 3B; W = 0.9741; *p* = 0.6369). Two-way ANOVA revealed a significant effect of voluntary running exercise (Figure 3B; F_1,27_ = 13.3, *p* = 0.0011) on promoting motor coordination in the vehicle-treated runners (Figure 3B; Tukey’s post-hoc: *p* = 0.0259 vs. Vehicle + Non-runners), though there was no main effect of interaction (Figure 3B; F_1,27_ = 0.08, *p* = 0.7787). In contrast, running failed to restore sorafenib-impaired motor coordination in vehicle-treated mice (Figure 3B; Tukey’s post-hoc: *p* = 0.0253 vs. Vehicle + Runners; F_1,27_ = 16.07, *p* = 0.0004).

In OFT, the ANOVA residuals in the distance travelled (Figure 3C; 5 min: W = 0.9588; *p* = 0.3063; 15 min: W = 0.9439, *p* = 0.1268) and mean speed (Figure 3E; W = 0.9646, *p* = 0.3468) were normally distributed. Two-way ANOVA revealed the significant effect of sorafenib on the total distance travelled (Figure 3C; F_1,25_ = 14.00, *p* = 0.0009) and the mean speed (Figure 3E; F_1,29_ = 5.06, *p* = 0.0322) in the first 5 min, but not the effects of voluntary running (Figure 3C; F_1,25_ = 1.2, *p* = 0.2829; Figure 3E; F_1,29_ = 0.34, *p* = 0.5590) and interactions (Figure 3C; F_1,25_ = 0.61, *p* = 0.4437; Figure 3E; F_1,25_ = 1.19, *p* = 0.2830). Sorafenib-treated runners travelled shorter distances than their vehicle-treated counterparts (Figure 3C; Tukey’s post-hoc: *p* = 0.0208 vs. Vehicle + Runners) in the first 5 min of OFT. We also measured a total locomotor activity total of 15 min, confirming sorafenib administration significantly reduced locomotor activity in both runners (Figure 3D; Tukey’s post-hoc: *p* = 0.0014 vs. Vehicle + Runners) and non-runners (Figure 3D; Tukey’s post-hoc: *p* = 0.0344 vs. Vehicle + Non-runners; effect of sorafenib: F_1,25_ = 25.95, *p* < 0.0001), which was coherent with the reduced daily home-cage wheel running activities (Figure 1D).

### 3.4. Neither Sorafenib Administration nor Voluntary Wheel Running Exercise Resulted in Any Changes in Muscle Weight or Strength

We further examined whether skeletal muscle mass and muscle strength contributed to sorafenib-impaired rotarod learning and reduced locomotor activity. Regardless of the body weight loss, sorafenib administration did not affect the overall muscle mass in the lower limb muscles compared to vehicle-treated non-runners (Appendix A; Tukey’s post-hoc: *p* > 0.05 vs. Vehicle + Non-runners). Specifically, sorafenib did not affect the masses of lower limb muscles, including gastrocnemius (Appendix A), soleus (Appendix A), plantar flexor (Appendix A), tibialis anterior (Appendix A), extensor digitorum longus (Appendix A), and plantar extensor (Appendix A) compared to vehicle-treated non-runners (Appendix A; *p* > 0.05 vs. Vehicle + Non-runners). As well as this, sorafenib administration did not affect the grip strengths of all four limbs (Appendix A; Tukey’s post-hoc: *p* > 0.05 vs. Vehicle + Non-runners) or the forelimbs alone (Appendix A; Tukey’s post-hoc: *p* > 0.05 vs. Vehicle + Non-runners), nor the latency to fall from the inverted grid (Appendix A; Tukey’s post-hoc: *p* > 0.05 vs. Vehicle + Non-runners).

### 3.5. Voluntary Wheel Running Increased Number of Immature Neurons in the Hippocampus

The ANOVA residuals in the proliferating cell counts were normally distributed (Figure 4A; ventral: W = 0.96, *p* = 0.35; Figure 4B; dorsal: W = 0.96, *p* = 0.53; and Figure 4C; whole: W = 0.94, *p* = 0.2092). Voluntary running exerted the significant main effect on proliferating cells (Ki-67^+^ cells) in DG (Figure 4A,D; ventral: F_1,20_ = 66.74, *p* < 0.0001; Figure 4B,E; dorsal: F_1,20_ = 29.04, *p* < 0.0001; and Figure 4C; whole: F_1,20_ = 66.06, *p* < 0.0001). Two-way ANOVA further indicated the significant effects of interaction (Figure 4A; ventral: F_1,20_ = 17.00, *p* = 0.0005; Figure 4B; dorsal: F_1,20_ = 15.22, *p* = 0.0009; Figure 4C; whole: F_1,20_ = 23.62, *p* < 0.0001) and sorafenib (Figure 4A; ventral: F_1,20_ = 15.34, *p* = 0.0009; Figure 4B; dorsal: F_1,20_ = 20.44, *p* = 0.0002; Figure 4C; whole: F_1,20_ = 26.42, *p* < 0.0001) on the number of proliferating cells in the DG.

Tukey’s post-hoc analyses showed that running significantly increased proliferating cells in the DG when compared to vehicle-treated non-runners (Figure 4A; ventral: *p* < 0.0001; Figure 4B; dorsal: *p* < 0.0001; Figure 4C; whole: *p* < 0.0001 vs. Vehicle + Non-runners). Sorafenib treatment did not significantly affect cell proliferation (Figure 4A; ventral: *p* > 0.05; Figure 4B; dorsal: *p* > 0.05; Figure 4C; whole: *p* > 0.05 vs. Vehicle + Non-runners). Running increased cell proliferation in the ventral DG (Figure 4A; *p* = 0.0440 vs. Sorafenib + Non-runners), but not the dorsal DG (Figure 4B; dorsal: *p* > 0.05 vs. Sorafenib + Non-runners; Figure 4C; whole: *p* > 0.05 vs. Sorafenib + Non-runners) in sorafenib-treated runners. The running-induced proliferative effect was more significant in vehicle-treated mice (Figure 4A; *p* < 0.0001 vs. Sorafenib + Runners).

On the other hand, sorafenib treatment did not affect the number of immature neurons (DCX^+^ cells) in the DG (Figure 5A,D; ventral: F_1,20_ = 0.3896, *p* = 0.5396; Figure 5B,E; dorsal: F_1,20_ = 0.28, *p* = 0.6022; Figure 5C; whole DG: F_1,20_ = 0.31, *p* = 0.5808). Voluntary exercise significantly increased the numbers of immature neurons (Figure 5A; ventral: F_1,20_ = 81.09, *p* < 0.0001; Figure 5B; dorsal: F_1,20_ = 74.08, *p* < 0.0001; and Figure 5C; whole: F_1,20_ = 100.1, *p* < 0.0001). The ANOVA residuals in the immature neuron cell counts were normally distributed (Figure 5A; ventral: W = 0.98, *p* = 0.85; Figure 5B; dorsal: W = 0.94, *p* = 0.2002; and Figure 5C; whole: W = 0.98, *p* = 0.82). Tukey’s post-hoc test showed that voluntary running increased the number of immature neurons in both vehicle-treated (Figure 5A; ventral: *p* < 0.0001; Figure 5B; dorsal: *p* < 0.0001; and Figure 5C; whole DG: *p* < 0.01 vs. Vehicle + Non-runners) and sorafenib-treated mice (Figure 5A; ventral: *p* < 0.0001; Figure 5B; dorsal: *p* < 0.0001; and Figure 5C; whole DG: *p* < 0.01 vs. Sorafenib + Non-runners).

## 4. Discussion

Physical exercise has been suggested as an adjuvant strategy to ameliorate cognitive decline and muscle weakness in cancer patients receiving chemotherapy [36,37,59]. Rodent studies have demonstrated the beneficial effects of physical exercise on chemotherapy-induced peripheral neuropathy [60] and cardiotoxicity [41]. Nonetheless, there is still a lack of literature illustrating the beneficial effects of physical exercise on mood-related behaviours, learning and memory and motor coordination as an adjuvant treatment to chemotherapy. In this study, we demonstrated that concomitant voluntary wheel running prevented sorafenib-associated motor skill impairment and anhedonia-like behaviour, although sorafenib-associated cognitive impairment in terms of spatial memory performance was not affected by this physical exercise protocol. In summary, the findings suggest that voluntary wheel running can be an effective adjunctive strategy to prevent sorafenib-induced impairments in motor learning and anhedonia for those undergoing chemotherapy.

Chemotherapy using VEGF receptor tyrosine kinase inhibitors, including sorafenib, is commonly associated with learning and memory impairments in cancer patients [29]. This study showed that sorafenib administration severely impaired spatial recognition memory even in healthy, non-tumour-bearing mice. Our findings are supported by Zhou et al., who demonstrated that sorafenib induced learning and memory impairments in the novel object recognition and Y-maze tasks [26]. However, concomitant voluntary exercise implemented in this study could not restore sorafenib-induced learning and memory deficits. A possible reason could be that sorafenib suppresses exercise-induced pro-cognitive effects by inhibiting VEGF signalling. A previous study has shown that neuronal-specific VEGF signalling inhibition impairs context-dependent fear memory retrieval, whereas subsequent disinhibition of VEGF signalling could not reverse the deficit [61]. Since VEGF is one of the peripheral hormones that mediate the pro-cognitive effects of physical exercise [62], voluntary wheel running could be ineffective in sorafenib-treated mice due to impaired VEGF signalling.

Adult hippocampal neurogenesis is essential to maintain learning and memory and modulate affective behaviours, whereas its impairment has been suggested as a potential mechanism contributing to chemotherapy-associated cognitive impairments [63]. Administrations of chemotherapeutic agents have been reported to interfere with adult hippocampal neurogenesis [64,65,66], which is associated with cognitive impairment. Despite the spatial working memory impairment, sorafenib administration did not affect adult neurogenesis in this study per se, although the physical exercise-associated increase in cell proliferation was obliterated in sorafenib-treated mice. Nonetheless, increased neuronal differentiation as a result of physical exercise was unaltered. A previous study has illustrated that promoting adult neurogenesis in the DG can confer stress resilience by reducing the neural excitability of the stress-responsive neurons [67], which can be one of the underlying mechanisms by which exercise could reduce depression-like behaviour in the forced swim test in both sorafenib- and vehicle-treated mice.

In this study, concomitant voluntary running exercise specifically improved anhedonia-like behaviour and motor skill acquisition in sorafenib-treated mice. It is known that dopaminergic signalling regulates both hedonic behaviour [68,69] and motor learning [70,71,72,73], while physical exercise can activate dopaminergic transmission. Adjuvant exercise paradigms can possibly restore sucrose preference by activating dopaminergic signalling [74]. Besides, the dopaminergic system can modulate motor plasticity [73]. Activation of VTA dopaminergic projection to M1 is necessary for successful motor skill acquisition [75,76], which can be blunted by pharmacological blockades of dopaminergic activation in M1 [70]. Voluntary running may improve hedonic preference and reinforce motor learning through dopamine signalling to counteract sorafenib-induced anhedonia and motor learning deficits.

The beneficial effects of voluntary running exercise on sorafenib-treated runners were limited to motor learning and did not affect motor coordination when assessed by the accelerated rotarod [77]. Impaired motor coordination and reduced locomotor activities in sorafenib-treated mice seemed independent of grip strength and skeletal muscle masses of the hindlimbs in all treatment groups. In fact, eight-week daily treadmill training could not improve speed and endurance in skeletal myofiber-specific VEGF-deficient mice, independent of skeletal muscle mass. Treadmill training-induced VEGF expressions in the hindlimb muscles, angiogenesis around the muscles, and expressions of oxidative enzymes were suppressed in these mutant mice [78]. This condition is known as exercise intolerance. Concertedly, these factors might explain why voluntary wheel running exercise could improve motor skill acquisition but not motor coordination in sorafenib-treated runners.

In this study, neither voluntary running exercise nor sorafenib administration changed grip strength and wire-hanging performance compared to vehicle-treated non-runners. Hout et al. [79] have reported that sorafenib treatment impairs function and structure in the skeletal muscles. Since the administration dosage and route of delivery are the same in both studies, it is possible that treatment duration affects the effects of sorafenib on muscle wasting and skeletal muscle strength, given that 28-day sorafenib treatment was applied in our study, while 42-day treatment was applied by Hout et al. The duration of voluntary wheel running may also partly explain why there were no differences in grip strength and wire-hanging performance between vehicle-treated runners and non-runners. Previous studies have illustrated that seven- to eight-week voluntary wheel running exercise [80,81] can improve grip strength in naive wild-type mice, whereas, in our study, voluntary running regimens lasted for only four weeks. Hence, the duration of chemotherapy administration and voluntary wheel-running exercise exposure may contribute to these discrepancies. Still, our results suggest that the onsets of cognitive impairment, reduced hedonic behaviours, and impaired motor learning might precede muscle wasting and motor dysfunction after chemotherapy. In this study, the beneficial effects of concomitant voluntary exercise on mood regulation and motor learning have shed light on the importance of early physical exercise intervention on chemotherapy-induced cognitive, mood, and motor dysregulations.

Sorafenib is prescribed to late-stage cancer patients in clinical practice. The use of non-tumour-bearing mice in this study might not entirely reflect the actual effect of voluntary wheel running exercise in clinical conditions. It has been reported that chemotherapy adversely affects cognitive performance and adult hippocampal neurogenesis in tumour-bearing mice [82,83,84]. Future studies should adopt the tumour-bearing mouse model with chemotherapy treatment to examine the neuroprotective effects of physical exercise for translational implication.

Above all, only voluntary running, but not the forced treadmill running paradigm, was used in this study. Since paired housing with a shared running wheel was used in our study, this limited the measurement of individual running activity for correlation analysis; and thus, we neglected the individual variability of exercise intensity. Future investigations may adopt forced treadmill exercise or single housing conditions to further examine the effects of physical running activity on reversing chemotherapy-associated negative impacts on the brain.

## 5. Conclusions

Our results demonstrated that concomitant voluntary exercise during chemotherapy was effective in improving anhedonia-like behaviour and motor skills acquisition in sorafenib-treated mice, emphasising the beneficial effects of early physical exercise intervention during chemotherapy. Our findings warrant further investigations on the interacting mechanisms between chemotherapy and physical exercise to identify efficacious physical exercise protocols to counteract chemotherapy-associated cognitive and motor declines.

## Figures and Tables

**Figure 1 ijerph-20-05371-f001:**
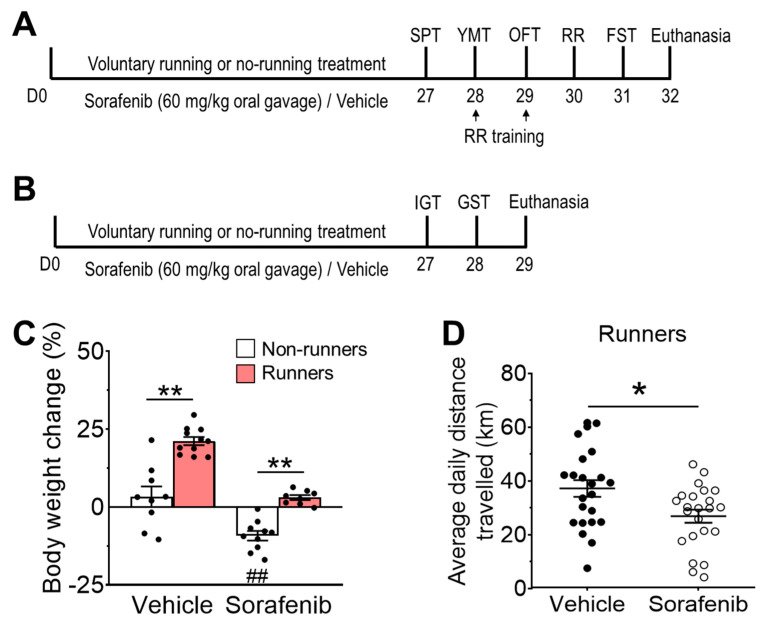
Voluntary running exercise attenuated sorafenib-induced weight loss. (**A**) Mice were housed with (runners) or without (non-runners) running wheels and were treated with sorafenib (60 mg/kg, oral gavage) or vehicle for 28 days continuously. They were then assessed by sucrose preference test (SPT), Y-maze test (YMT), open field test (OFT), rotarod test (RR), and forced swim test (FST). Fixed tissues were collected for histological analyses at the experimental endpoint. (**B**) In the second cohort, mice were submitted to muscle strength tests, including the inverted grip test (IGT) and grip strength test (GST) after the same treatment. Fresh skeletal muscles were collected at the experimental endpoint. (**C**) Sorafenib administration significantly reduced body weight. Concomitant voluntary exercise promoted body weight gain in vehicle-treated runners and restored body weight loss in sorafenib-treated runners (Tukey’s post-hoc test: ** *p* < 0.005 vs. Vehicle + Non-runners & Vehicle + Runners; ^##^ *p* < 0.005 vs. Vehicle + Non-runners). Results were expressed as mean ± SEM. *n* = 8–11 animals per group. (**D**) The average running distance was significantly reduced in sorafenib-treated runners (unpaired *t*-test: ** *p* < 0.005). * *p* < 0.05.

**Figure 2 ijerph-20-05371-f002:**
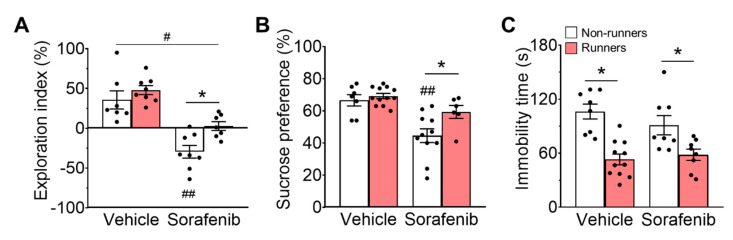
Sorafenib administration impaired spatial memory and induced anhedonia-like behavior. (**A**) Sorafenib administration impaired spatial working memory in non-runners (Tukey’s post hoc: ^##^ *p* < 0.005 vs. Vehicle + Non-runners, * *p* < 0.05). Running failed to improve cognitive impairment in sorafenib-treated mice since they did not show any preferences for novel over familiar arms. Voluntary running could not restore sorafenib-induced cognitive impairment (Tukey’s post hoc: ^#^ *p* < 0.05 vs. Vehicle + Non-runners). (**B**) Sorafenib administration induced anhedonia-like behaviour as showed with a significant reduction in sucrose preference (Tukey’s post hoc: ^##^ *p* < 0.005 vs. Vehicle + Non-runners), whereas voluntary running showed the opposite effects (Tukey’s post hoc: * *p* < 0.05 vs. Sorafenib + Non-runners). (**C**) Sorafenib administration did not affect the immobility time spent in FST, while voluntary running reduced immobility time (Tukey’s post hoc: * *p* < 0.05 vs. Non-runners). Results were expressed as mean ± SEM. *n* = 7–12 animals per group.

**Figure 3 ijerph-20-05371-f003:**
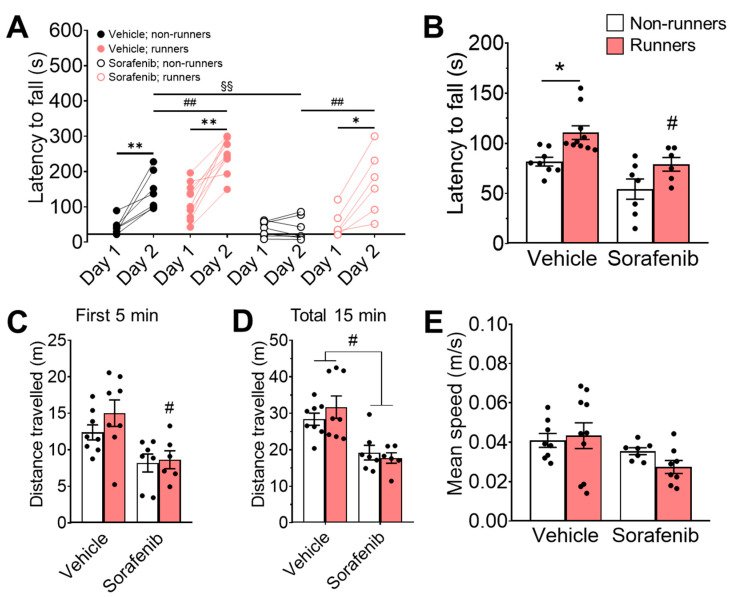
Exercise ameliorated sorafenib-impaired motor learning. (**A**) Vehicle-treated and non-runners showed improved motor learning on the second day on the constant-speed rotarod (Tukey’s post-hoc test: ** *p* < 0.005 vs. Day 1: Sorafenib + Runners & Non-runners) ^##^ *p* < 0.005. Sorafenib administration impaired rotarod learning (Tukey’s post-hoc test: ^§§^ *p* < 0.005 vs. Day 2: Vehicle + Non-runners), which could be restored by concomitant voluntary wheel running exercise (Tukey’s post-hoc test: * *p* < 0.05 vs. Day 1: Sorafenib + Runners). (**B**) Voluntary wheel running exercise improved motor coordination of vehicle-treated runners on the accelerated rotarod with increased latency to fall (Tukey’s post-hoc test: * *p* < 0.05 vs. Vehicle + Non-runners). Exercise-associated motor coordination improvements were hindered by sorafenib administration in sorafenib-treated runners (Tukey’s post-hoc test: ^#^ *p* < 0.05 vs. Sorafenib + Runners). (**C**,**D**) There was a significant reduction in distance travelled in (**C**) sorafenib-treated runners during the first 5 min (Tukey’s post-hoc test: # *p* < 0.05 vs. Vehicle-treated + Non-runners) and (**D**) in both sorafenib-treated runners and non-runners during the total 15 min in the open field (Tukey’s post-hoc test: # *p* < 0.05 vs. Vehicle-treated + Runners & Vehicle-treated + Non-runners). (**E**) Neither voluntary wheel running exercise nor sorafenib administration affected mean speed. Results were expressed as mean ± SEM. *n* = 7–12 per group.

**Figure 4 ijerph-20-05371-f004:**
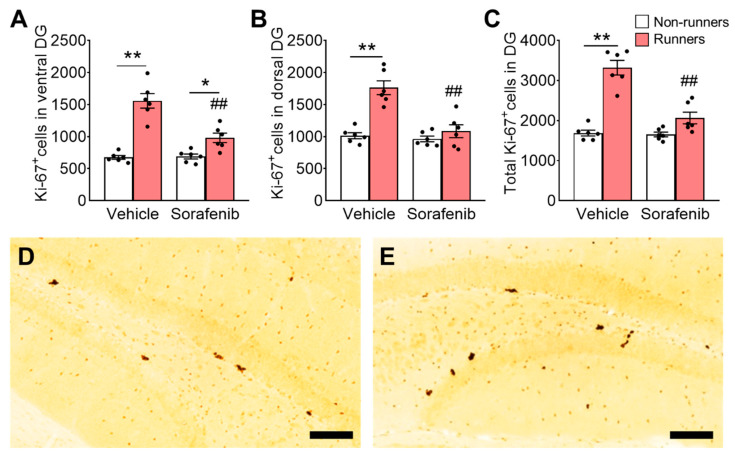
Voluntary running failed to increase cell proliferation in DG of sorafenib-treated mice. (**A**) Voluntary running exercise increased proliferating progenitors in the ventral DG of vehicle-treated mice (Tukey’s post hoc: ** *p* < 0.005 vs. Vehicle + Non-runners). Sorafenib did not affect the proliferating progenitor number. Although voluntary exercise increased the proliferating cell counts in sorafenib-treated mice (Tukey’s post hoc: * *p* < 0.05 vs. Sorafenib + Non-runners), the exercise-induced proliferation was significantly lower (Tukey’s post hoc: ^##^ *p* < 0.005 vs. Sorafenib + Non-runners) in the ventral DG. (**B**,**C**) Similarly, concomitant voluntary exercise promoted cell proliferation in the (**B**) dorsal and (**C**) whole DG in vehicle-treated runners (Tukey’s post hoc: ** *p* < 0.005 vs. Vehicle + Non-runners), but not in sorafenib treated mice (Tukey’s post hoc: ^##^ *p* < 0.005 vs. Vehicle + Runners). (**D**,**E**) Representative images of Ki-67 positive cells in the (**D**) ventral and (**E**) dorsal DG (Scale bars, 100 μm in 200×). Results were expressed as mean ± SEM. *n* = 6 per group.

**Figure 5 ijerph-20-05371-f005:**
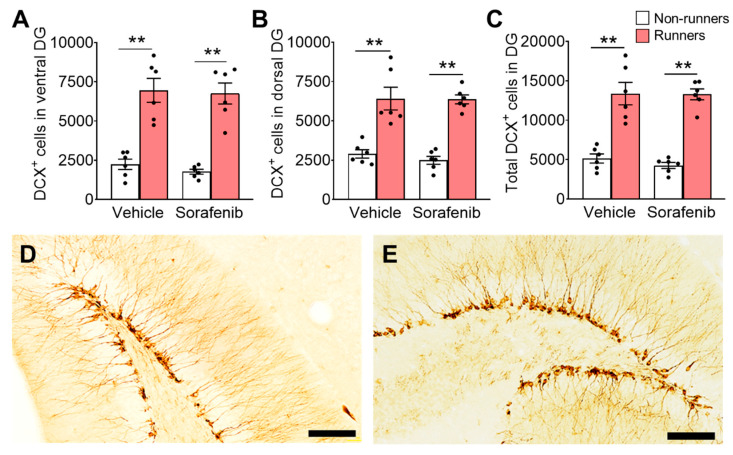
Voluntary running promoted immature neuron numbers in DG, but sorafenib showed no effect. (**A**–**C**) Sorafenib administration did not affect immature neuron number in DG, while runners increased immature neurons in the (**A**) ventral, (**B**) dorsal, and (**C**) whole DG of sorafenib-treated and vehicle-treated mice (Tukey’s post hoc: ** *p* < 0.005 vs. Non-runners). (**D**,**E**) Representative images of DCX positive cells in the (**D**) ventral and (**E**) dorsal DG (Scale bars, 100 μm in 200×). Results were expressed as mean ± SEM. *n* = 6 per group.

## Data Availability

Data may be made available upon sending requests to the correspondence.

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
