# Peer review of "Effects of Voluntary Wheel Running Exercise on Chemotherapy-Impaired Cognitive and Motor Performance in Mice"

_ijerph, 2023, doi:10.3390/ijerph20075371_

Round 1
Reviewer 1 Report
The paper entitled " Effects of voluntary wheel running exercise on chemotherapy- 2
impaired cognitive and motor performance in mice " is an experimental study that aimed to investigate the efficacy of physical exercise as a non-pharmacological intervention to reverse these adverse side effects of chemotherapy.
The manuscript is well written and well supported by consistent references. The conclusions are largely sound and improve the existing knowledge.
Some suggestions:
- The authors incorrectly spelled the p-value with a capital "P" throughout the manuscript
- It is not clear how many animals were used in this experimental study
- Line 297: the authors wrote “Figure Legends” twice
- The results paragraph is written in italics, perhaps this is a careless error?
- The authors do not discuss the limitations of the study; it is important to include a paragraph discussing the limitations of this experimental study
Author Response
Responses to Reviewer #1
Thank you so much for your positive comments. We have improved the contents as suggested by the reviewer accordingly. Here are the point-to-point response to the major suggestions.
Point 1: The authors incorrectly spelled the p-value with a capital "P" throughout the manuscript
Response: Thank you very much for your comment. We have amended them accordingly.
Point 2: It is not clear how many animals were used in this experimental study
Response: Thank you very much for your observation. We have specified the number of mice involved in each experiment in the figure legend.
Point 3: Line 297: the authors wrote “Figure Legends” twice
Response: We have corrected it accordingly.
Point 4: The results paragraph is written in italics, perhaps this is a careless error?
Response: Thank you for pointing out the mistakes, we have formatted the style in this revised manuscript.
Point 5: The authors do not discuss the limitations of the study; it is important to include a paragraph discussing the limitations of this experimental study.
Response: Thank you very much for your suggestion. We have included a paragraph to describe the limitations in this study in the last paragraph of the Discussion session (line number 504-518).
Sorafenib is prescribed to late-stage cancer patients in clinical practice. The use of non-tumour-bearing mice in this study might not fully reflect the actual effect of voluntary wheel running exercise in clinical condition. It has been reported that chemotherapy has adverse effects on cognitive performance and adult hippocampal neurogenesis in tu-mour-bearing mice [80-82]. Future study should adopt the tumour-bearing mouse model with chemotherapy treatment to examine neuroprotective effects of physical exercise for translational implication.
Besides, voluntary running, but not the forced treadmill running paradigm was used in this study. Since paired housing with a shared running wheel was used in our study, limiting the measurement of individual running activity for correlation analysis, potential exercise intensity variability among individual mice may have been missed in our study. Future investigations may adopt forced treadmill exercise or single housing condition in order to further examine the effects of physical running activity on restoring chemo-therapy-associated negative impact on the brain.
Reviewer 2 Report
ID#: IJERPH-2255640
Manuscript title:
Effects of voluntary wheel running exercise on chemotherapy-2 impaired cognitive and motor performance in mice
Overview
The aim of this study was to determine physical exercise as a non-pharmacological intervention to reverse the adverse side effects of chemotherapy. Results showed that physical exercise (voluntary wheel running) prevented body weight loss, improved motor skill acquisition, and reduced anhedonia-like behaviour in mice treated with the multi-kinase inhibitor sorafenib. Running increased the number of proliferating cells and immature neurons in the hippocampal dentate gyrus. The authors concluded that physical exercise rehabilitation effectively prevents chemotherapy side effects regarding mood dysregulation and motor deficit.
Comments and suggestions
Title
The title was clear, concise, and reflected the study.
Abstract
The abstract reflects the study’s purpose, objective, methodology, main findings, and conclusion. However, the methodology needs to include information, and the summary of results does not show any data and statistical significance.
Please include how much physical exercise was done and how often in the abstract.
Please include mean ± SD and statistical significance (e.g., p values) in the abstract.
Introduction.
The introduction is clear, stating the research topic, research and objectives. However, the originality and novelty of the study and the literature gaps need to be further explored. The authors need to try to demonstrate why this study is original and novel. For example, what is the novelty of this study? What does your research add to the literature? Please include this information in the introduction. Hypotheses are missing in the introduction. What are your hypotheses? Please have your hypotheses included in the introduction.
Specific comments
Line 43. Please change the word subjects to patients.
Methods
The methods session presented are adequately aligned with the research question and objectives. However, some important information needs to be included.
What are the primary and secondary outcomes of the study?
Data collection is clearly described, but some parts need to provide more details. It would be interesting to explain the data analysis in more detail.
Statistical analysis is appropriate to the research design. However, average distribution analysis was not reported.
Specific comments
Line 73. How many mice are in each group, and how did you decide the sample size of your study? Did you perform any prior power calculations to determine your sample size? Please report this information.
Line 88. Voluntary wheel running exercise.
Although you mentioned that the voluntary wheel running was previously described (reference #49), you need to briefly provide relevant details about it. You should report frequency (how often?), duration of the physical exercise program (How long?) and training session (How long?), and intensity (How hard?).
Line 117. Open Field Test (OFT)
Although you mentioned that the OFT was previously described (reference #50). Please can you provide more details? How was data analyzed?
Line 178. Statistical analyses
Did you test for normal distribution? Please include this information in the Statistical analyses session.
Result
The results are described clearly and aligned with the research objectives and hypotheses.
I found it interesting how you named the subtitles of your results session, emphasizing the main finding of that specific variables. The figures were well-designed. However, it would be best if you were more consistent on how to deliver this information. My suggestion is: first, include normality test results. Then, describe the main results of the statistical analysis, including the main effects and interactions (ANOVAs two and three-way), followed by the post-hoc descriptive results.
Specific comments
Post-hoc test information is missing in the results. Please report the post-hoc tests.
Lines 189-223
Please report the interactions immediately after the main effect, not at the end of the paragraph. For example, Two-way ANOVA revealed a significant effect of sorafenib on body weight change (Fig. 1C; F1,34 = 190 61.46, P < 0.0001). However, there was no interaction between sorafenib administration and voluntary running (Fig. 1C; F1,34 = 2.48, P = 0.1615)”. Then, you continue describing the main effects of the variables. Please apply these changes in the results session when it is necessary.
Line 245.
“However, there was a main effect of interaction (Fig. 3B; F1,27 = 0.08, P = 0.7787)”. Sorry, it is not clear to me. There was there a main effect or not? Your reported p-value of 0.7787. Please can you clarify?
Discussion
Statements are clear and concise. The authors presented the objectives and main findings at the beginning of the discussion. However, I missed a sentence explaining the study’s originality and novelty in the first paragraph. Results are discussed based on the literature, and interpretations are coherent with previous studies. Limitations of the study were not included in the discussion.
Specific comments
Line 380-382. This sentence should be introduced much earlier in the manuscript because it is linked with the novelty of this study. You should include information about this sentence in your introduction and the first paragraph of your discussion.
Are any limitations of the current study? Please include them and articulate how such limitations may or may not impact your results.
Conclusion
Conclusions align with the research question and objectives.
References
Please check if the references are following the journal instructions.
Author Response
Responses to Reviewer #2
Thank you so much for your constructive comments. We have improved the contents as suggested by the reviewer accordingly. Here are the point-to-point response to the major suggestions.
Point 1: The abstract reflects the study’s purpose, objective, methodology, main findings, and conclusion. However, the methodology needs to include information, and the summary of results does not show any data and statistical significance.
Response: We have amended them accordingly
Point 2: Please include how much physical exercise was done and how often in the abstract.
Response: We acknowledged that this is the limitation of our study that we did not consider the exercise amount. But we have clarified it in the abstract line 21-22 as below:
Mice can freely access the running wheel anytime during sorafenib or vehicle treatment period.
Point 3: Please include mean ± SD and statistical significance (e.g., p values) in the abstract.
Response: We have amended them accordingly.
Point 4: The introduction is clear, stating the research topic, research and objectives. However, the originality and novelty of the study and the literature gaps need to be further explored. The authors need to try to demonstrate why this study is original and novel. For example, what is the novelty of this study? What does your research add to the literature? Please include this information in the introduction. Hypotheses are missing in the introduction. What are your hypotheses? Please have your hypotheses included in the introduction.
Response: Thank you very much for your suggestion. We have refined the last paragraph of the introduction to highlight our research novelty and hypothesis from line 74-79.
“Physical exercise could be an adjuvant non-pharmacological intervention for improving chemotherapy-induced impairments in mood regulation, learning and memory and motor performance. In this study, we therefore hypothesize that (1) concomitant voluntary running exercise not only prevents chemotherapy-induced muscle wasting, but also improves learning and memory impairments; and (2) changes of hippocampal neurogenesis are involved in behavioural improvement following exercise intervention in mice received chemotherapy. “
Point 5: Line 43. Please change the word subjects to patients.
Response: We have amended it accordingly.
Point 6: The methods session presented are adequately aligned with the research question and objectives. However, some important information needs to be included.
What are the primary and secondary outcomes of the study?
Response: Primary outcome was the changes in the behavioural assessments related to learning and memory and depression-like behaviours in sorafenib-treated mice receiving concomitant voluntary running exercise treatment. Secondary outcomes included the underlying changes in adult hippocampal neurogenesis, motor performances, and skeletal muscle weight changes.
Point 7: Data collection is clearly described, but some parts need to provide more details. It would be interesting to explain the data analysis in more detail. Statistical analysis is appropriate to the research design. However, average distribution analysis was not reported.
Response: We have amended it accordingly in result section.
The ANOVA residual of body weight change was normally distributed (Fig. 1C; W = 0.9561; p = 0.1419). Line 216-217.
The ANOVA residual of exploration index in the Y-maze test was normally distributed (Fig. 2A; W = 0.95; p = 0.1757). Line 233-234.
The ANOVA residuals in SPT (Fig. 2B; W = 0.9580; p = 0.1866) fell into normality. Line 246
Although the ANOVA residuals in the FST suggested non-normal distribution, a post hoc analysis using the Mann Whitney comparison test confirmed significant effects of running for vehicle- (Mann-Whitney U = 3, p = 0.0002 vs. Non-runners) and sorafenib-treated (Mann-Whitney U = 12, p = 0.0379 vs. Non-runners) mice. Line 259-262.
The ANOVA residuals in the latency to fall were normally distributed (Fig. 3B; W = 0.9839; p = 0.6359). Line 265-266
The ANOVA residuals in the latency to fall were normally distributed (Fig. 3B; W = 0.9741; p = 0.6369). Line 283-284
In OFT, the ANOVA residuals in distance travelled (Fig. 3C; 5 min: W = 0.9588; p = 0.3063; 15 min: W = 0.9439, p = 0.1268) and mean speed (Fig. 3E; W = 0.9646, p = 0.3468) were normally distributed. Line 290-292
The ANOVA residuals in the proliferating cell counts were normally distributed (Fig. 4A; ventral: W = 0.96, p = 0.35; Fig. 4B; dorsal: W = 0.96, p = 0.53; and Fig. 4C; whole: W = 0.94, p = 0.2092). Line 322-324
The ANOVA residuals in the immature neuron cell counts were normally distributed (Fig. 5A; ventral: W = 0.98, p = 0.85; Fig. 5B; dorsal: W = 0.94, p = 0.2002; and Fig. 5C; whole: W = 0.98, p = 0.82). Line 346-348
Point 8: Line 73. How many mice are in each group, and how did you decide the sample size of your study? Did you perform any prior power calculations to determine your sample size? Please report this information.
Response: We have provided the calculation in section 2.1 and Line 90-92.
Based on our pilot experiment investigating the impact of sorafenib treatment on cognition (Cohen's d = 1.60), we estimated that eight animals per group would be necessary to achieve a study power of 80%, considering a level of significance of 5%.
Point 9: Line 88. Voluntary wheel running exercise.
Although you mentioned that the voluntary wheel running was previously described (reference #49), you need to briefly provide relevant details about it. You should report frequency (how often?), duration of the physical exercise program (How long?) and training session (How long?), and intensity (How hard?).
Response: In this voluntary exercise running paradigm, the intensity, frequency, and duration of running per day are not restricted. A running wheel is mounted in the home cage in which two mice can freely access the wheels anytime for 28 days before behavioural assessments. We have further elaborated our protocol in Line 102-103.
Two runner mice shared a cage equipped with a running wheel and could freely access the wheel 24 h per day over 28 days of treatment.
Point 10: Line 117. Open Field Test (OFT)
Although you mentioned that the OFT was previously described (reference #50). Please can you provide more details? How was data analyzed?
Response: The analyses of distance travelled and mean speed were automatically generated by ANYMAZE software from our videotapes. We have further clarified the procedures in Line 134-138.
Locomotor activity in the open field box was video-taped. Distance travelled and mean speed over the first 5 min and the total 15 min were automatically detected and calculated by ANYMAZE software (Stoelting Co., IL, USA). The arena was cleaned with 70% ethanol and water between tests to eliminate olfactory cues.
Point 11: Line 178. Statistical analyses
Did you test for normal distribution? Please include this information in the Statistical analyses session.
Response: In this revision, accordingly in Statistical analyses section line 204-206.
Datasets with p ≥ 0.05 in the Shapiro–Wilk test for ANOVA residuals were considered not deviated from normality.
Point 12: Result
The results are described clearly and aligned with the research objectives and hypotheses.
I found it interesting how you named the subtitles of your results session, emphasizing the main finding of that specific variables. The figures were well-designed. However, it would be best if you were more consistent on how to deliver this information. My suggestion is: first, include normality test results. Then, describe the main results of the statistical analysis, including the main effects and interactions (ANOVAs two and three-way), followed by the post-hoc descriptive results.
Specific comments
Post-hoc test information is missing in the results. Please report the post-hoc tests.
Response: We employed Tukey’s post-hoc test in all our analyses. We have specified the tests in the bracket indicating p-values.
Point 13: Lines 189-223
Please report the interactions immediately after the main effect, not at the end of the paragraph. For example, Two-way ANOVA revealed a significant effect of sorafenib on body weight change (Fig. 1C; F1,34 = 190 61.46, P < 0.0001). However, there was no interaction between sorafenib administration and voluntary running (Fig. 1C; F1,34 = 2.48, P = 0.1615)”. Then, you continue describing the main effects of the variables. Please apply these changes in the results session when it is necessary.
Response: We have made changes in the result session to deliver the statistics efficiently.
Point 14: Line 245.
“However, there was a main effect of interaction (Fig. 3B; F1,27 = 0.08, P = 0.7787)”. Sorry, it is not clear to me. There was there a main effect or not? Your reported p-value of 0.7787. Please can you clarify?
Response: We apologize for the confusions. We have made amendments accordingly in line 283-284.
The ANOVA residuals in the latency to fall were normally distributed (Fig. 3B; W = 0.9741; p = 0.6369).
Point 15: Line 380-382. This sentence should be introduced much earlier in the manuscript because it is linked with the novelty of this study. You should include information about this sentence in your introduction and the first paragraph of your discussion.
Response: We have refined the discussion as you suggested to highlight the novelty of this study.
Point 16: Are any limitations of the current study? Please include them and articulate how such limitations may or may not impact your results.
Response: We have enlisted the limitations in the current study from line number 504-518.
Sorafenib is prescribed to late-stage cancer patients in clinical practice. The use of non-tumour-bearing mice in this study might not fully reflect the actual effect of voluntary wheel running exercise in clinical condition. It has been reported that chemotherapy has adverse effects on cognitive performance and adult hippocampal neurogenesis in tu-mour-bearing mice [80-82]. Future study should adopt the tumour-bearing mouse model with chemotherapy treatment to examine neuroprotective effects of physical exercise for translational implication.
Besides, voluntary running, but not the forced treadmill running paradigm was used in this study. Since paired housing with a shared running wheel was used in our study, limiting the measurement of individual running activity for correlation analysis, potential exercise intensity variability among individual mice may have been missed in our study. Future investigations may adopt forced treadmill exercise or single housing condition in order to further examine the effects of
Point 17: References
Response: We have amended it accordingly.
Reviewer 3 Report
1. Line 64: What does "in contract" mean in this sentence?
2. Line 71: The authors should provide hypothesis for this study.
3. Line 78: What's the definition of the first cohort and the second cohort? The authors should explain why did these mice have to be seperated into two cohort instead of all received both kinds of examination. The number of mice used in this study should also be mentioned in this section, and the authors should provide the results of sample size calculation here.
4. Line 88: What's the influence of the total distance of wheel running exercise on the outcome measurements of this study? Is there any correlation between the time spent on wheel running and your outcome measurements? The authors just simplified the intervention as exercise and non-exercise, and ignored the effect of the dosage of exercise.
5. Line 108: Why does sucrose preference relate to anhedonia-like behaviour? The authors should briefly describe the underlying reason instead of just providing reference.
6. Line 121: Again, what's the relationship between the reluctance to swim and depression? The authors should briefly describe the story.
7. Line 177: Why didn't the authors examine the ratio of type I and type II muscle fibers and compare them between the exercise and non-exercise mice??
8. Line 180: How could the authors use paired t test to examine the relationship of two independent scienarios? Paried t test should be used to examine between the data "before" and "after" intervention. Furthermore, is the data normally distributed? The authors should use Mann Whitney U test instead of t test if the data is not normally distributed.
9. In the end of every figure legend, the authors revealed the number of mice per group, some are 8-11 and some are 7-12. The authors should provide the definite number of mice for each test intead of just providing a range. Furthermore, is the animal number enough to achieve sufficient statistical power?
Author Response
Responses to Reviewer #3
Thank you so much for your positive comments. We have improved the contents as suggested by the reviewer accordingly. Here are the point-to-point response to the major suggestions.
Authors’ response
Point 1: Line 64: What does "in contract" mean in this sentence?
Response: Thank you for pointing out the mistake, here we mean ‘in contrast’ and we have made the amendment.
Point 2: Line 71: The authors should provide hypothesis for this study.
Response: We have amended our sentences to provide a clear from line 74 to 79
Physical exercise could be an adjuvant non-pharmacological intervention for improving chemotherapy-induced impairments in mood regulation, learning and memory and motor performance. In this study, we therefore hypothesize that (1) concomitant voluntary running exercise not only prevents chemotherapy-induced muscle wasting, but also improves learning and memory impairments; and (2) changes of hippocampal neurogenesis are involved in behavioural improvement following exercise intervention in mice received chemotherapy.
Point 3: Line 78: What's the definition of the first cohort and the second cohort? The authors should explain why did these mice have to be seperated into two cohort instead of all received both kinds of examination. The number of mice used in this study should also be mentioned in this section, and the authors should provide the results of sample size calculation here.
Response: Two separate batches of animals were used for two difference experiments. We first intended to examine behavioural changes related to learning and memory performance and emotion-related behaviours after sorafenib administration and exercise treatment. We then aimed to study the underlying histological changes in the hippocampus, we collected fixed brain tissues as soon as the behavioural tests were completed since the sorafenib administration and exercise treatment were discontinued before the beginning of the test period.
We then found that motor learning was altered upon sorafenib treatment. Therefore, we were interested in further our investigations on whether the behavioural outcomes might be linked to motor deficits. Fresh brain and muscle tissues were collected in this cohort.
In this revision, we have mentioned the number of mice used and provided the sample size in line 90-92.
Based on our pilot experiment investigating the impact of sorafenib treatment on cog-nition (Cohen's d = 1.60), we estimated that eight animals per group would be necessary to achieve a study power of 80%, considering a level of significance of 5%.
Point 4: Line 88: What's the influence of the total distance of wheel running exercise on the outcome measurements of this study? Is there any correlation between the time spent on wheel running and your outcome measurements? The authors just simplified the intervention as exercise and non-exercise, and ignored the effect of the dosage of exercise.
Response: We admit that this is a limitation of a voluntary paradigm and in our study. Since these mice are subjected to behavioural assessments after treatment period, we housed 2 mice in each cage equipped with a shared running wheel to reduce social isolation stress. The running activity recorded was a total activity of two animals. Therefore, we are unable to correlate behavioural outcomes and exercise intensity and duration of an individual animal I this study. We have discussed this study design as an experimental limitation in discussion as below (line 504 to 518):
Sorafenib is prescribed to late-stage cancer patients in clinical practice. The use of non-tumour-bearing mice in this study might not fully reflect the actual effect of voluntary wheel running exercise in clinical condition. It has been reported that chemotherapy has adverse effects on cognitive performance and adult hippocampal neurogenesis in tu-mour-bearing mice [80-82]. Future study should adopt the tumour-bearing mouse model with chemotherapy treatment to examine neuroprotective effects of physical exercise for translational implication.
Besides, voluntary running, but not the forced treadmill running paradigm was used in this study. Since paired housing with a shared running wheel was used in our study, limiting the measurement of individual running activity for correlation analysis, potential exercise intensity variability among individual mice may have been missed in our study. Future investigations may adopt forced treadmill exercise or single housing condition in order to further examine the effects of physical running activity on restoring chemo-therapy-associated negative impact on the brain.
Point 5: Line 108: Why does sucrose preference relate to anhedonia-like behaviour? The authors should briefly describe the underlying reason instead of just providing reference.
Response: Consuming sweet foods or solutions is an innate preference of naïve, non-stressed rodents (Sclafani Brain Res Bull. 1991 doi: 10.1016/0361-9230(91)90129-8). In SPT, sucrose solution is presented as a reward. Conversely, anhedonia refers to the reduced ability to experience pleasure (Gorwood Dialogues Clin Neurosci. 2008 doi: 10.31887/DCNS.2008.10.3/pgorwood), which is a core symptom of depression in humans. Rodent studies have demonstrated that mice reduce sweet solution consumption upon experiencing chronic stress (Liu et al Nat Protoc 2018 doi: 10.1038/s41596-018-0011-z.), while this reduction can be reversed by antidepressant treatments (Yang et al Transl Psychiatry 2015 doi: 10.1038/tp.2015.136) or non-pharmacological treatments like physical exercise (Xiao et al Transl Psychiatry 2021 doi: 10.1038/s41398-021-01571-9). We have further clarified the procedure from Line 123 to 125.
Rodents have an innate preference for sweet-tasting food and drinks [51], whereas stressors can induce anhedonia-like behaviour as indicated by a reduced tendency to seek palatable rewards [52].
Point 6: Line 121: Again, what's the relationship between the reluctance to swim and depression? The authors should briefly describe the story.
Response: In FST, swimming/climbing activity in the water cylinder is an active coping strategy in an attempt to escape after exposing to a stressful environment (Commons et al ACS Chem Neurosci 2017 doi: 10.1021/acschemneuro.7b00042). Conversely, immobility is an indication of behavioral despair which resembles clinical conditions in which depressed human subjects present passive coping. Rodent studies have demonstrated that immobility time increased in mice experienced chronic stress (Sequeira-Cordero et al Sci Rep 2019 DOI: 10.1038/s41598-019-53624-1), while antidepressant treatments and exercise treatment (Yau et al Proc Natl Acad Sci U S A. 2014 doi: 10.1073/pnas.1415219111.) can reduce immobility time spent in FST. We have further clarified the procedure from Line 142 to 146.
The test examines coping strategy adopted by the mice under an acute inescapable stress [53]. Struggling and swimming are active coping strategies to avoid negative outcome, such as drowning, whereas immobility is a passive coping strategy indicative of de-pression-like behaviour, as antidepressants tend to increase active in detriment of passive coping strategies.
Pont 7: Line 177: Why didn't the authors examine the ratio of type I and type II muscle fibers and compare them between the exercise and non-exercise mice??
Response: Since there were no observable differences in grip strength and wire hanging performance in all 4 groups, we did not carry out histological analyses on the skeletal muscle samples. Still, we believe that prolonged treatment may alter skeletal muscular functions with associated histological changes in the ratio of muscle fiber type as previously reported by Huot et al (Cancers, 2019, doi.org/10.3390/cancers11040571).
Point 8: Line 180: How could the authors use paired t test to examine the relationship of two independent scienarios? Paried t test should be used to examine between the data "before" and "after" intervention. Furthermore, is the data normally distributed? The authors should use Mann Whitney U test instead of t test if the data is not normally distributed.
Response: The running activities of vehicle-treated runners (W = 0.9644; p = 0.5579) and sorafenib-treated runners (W = 0.9434; p = 0.2123) were normally distributed according to Shapiro-Wilk test.
The dataset presents the average daily running activity of two groups of runners receiving either vehicle or sorafenib treatment. Each data point represents the average running activities of all cages of mice from the same treatment group (with 2 mice per cage) per day. Therefore, we employed the paired t-test to compare the average daily running activity of vehicle-treated runners versus sorafenib-treated runners on a day-to-day basis. We have formatted the graphical representation for a better understanding.
Point 9: In the end of every figure legend, the authors revealed the number of mice per group, some are 8-11 and some are 7-12. The authors should provide the definite number of mice for each test intead of just providing a range. Furthermore, is the animal number enough to achieve sufficient statistical power?
Response: We have specified the number of mice per group accordingly in figure legend. In this revision, we have mentioned the number of mice used and provided the sample size in line 90-92.
Based on our pilot experiment investigating the impact of sorafenib treatment on cog-nition (Cohen's d = 1.60), we estimated that eight animals per group would be necessary to achieve a study power of 80%, considering a level of significance of 5%.
Round 2
Reviewer 2 Report
The authors did a good job, including the suggestions.
Author Response
Responses to Reviewer #2
Thank you so much for your kind and positive feedbacks. we greatly appreciate your satisfaction with our responses and the valuable time you took to provide guidance.
Reviewer 3 Report
1. Line 74: Before describing your hypothesis, the aim of this study should be clearly stated, following by the hypothesis of this study.
2. Line 206: Paired t test can only be used in paired sets of data, each observation in one set has exactly one corresponding observation is another set, for example, pre- and post-test scores on the same person. The number of animals in each group is different, ranging from 7-12 or 8-11, which cannot be analyzed by paired t test. The authors should adopt student t test instead. Furthermore, why not use the same number of animals, for example, 10 mice in each group?
3. Line 492: It is interesting there were no observable differences in grip strength and wire hanging performance in all 4 groups in this study, which had different conclusion comparing to reference 79. The authors should compare the the experimental protocols in the two studies and at least try to explain the reason behind in this discussion section.
Author Response
Responses to Reviewer #3
Thank you so much for your constructive comments. We have improved the contents as suggested by the reviewer accordingly. Here are the point-to-point response to the major suggestions.
Point 1. Line 74: Before describing your hypothesis, the aim of this study should be clearly stated, following by the hypothesis of this study.
Response: We have improved our description in the revised manuscript From line 74 to 81 in the introduction
In this study, we aim to investigate whether chemotherapy impairs motor performance, mood regulation, learning and memory, which can be reversed by physical wheel running in a mouse model. We further examine whether behavioural changes are associated with changes of adult hippocampal neurogenesis. We hypothesize that voluntary running exercise prevents chemotherapy-induced muscle wasting and improves learning and memory impairments in association with changes of hippocampal neurogenesis.
Point 2. Line 206: Paired t test can only be used in paired sets of data, each observation in one set has exactly one corresponding observation is another set, for example, pre- and post-test scores on the same person. The number of animals in each group is different, ranging from 7-12 or 8-11, which cannot be analyzed by paired t test. The authors should adopt student t test instead. Furthermore, why not use the same number of animals, for example, 10 mice in each group?
Response: In revised manuscript, we have made amendments on the statistical test as in Line 226 and Line 246 as below:
An unpaired t-test with Welch’s correction
(Fig. 1D; t41.59 = 2.620, p = 0.0122)
We were unable to obtain a full batch of 40 mice and subdivide them into 4 groups at the beginning in Cohort 1. Therefore, we had to obtain batches of mice, in which the number of mice received varied from batch to batch, and we completed all the experiments by staggering the treatments and experiments.
Point 3. Line 492: It is interesting there were no observable differences in grip strength and wire hanging performance in all 4 groups in this study, which had different conclusion comparing to reference 79. The authors should compare the the experimental protocols in the two studies and at least try to explain the reason behind in this discussion section.
Response: We have addressed the discrepancy in the revised manuscript from line 531 to 544 in the discussion as below:
In this study, neither voluntary running exercise nor sorafenib administration changes grip strength and wire-hanging performance when compared with vehicle-treated non-runners. Hout et al [79] have reported that sorafenib treatment impairs function and structure in the skeletal muscles. Since the administration dosage and route of delivery are the same in both studies, it is possible that treatment duration affects the effects of sorafenib on muscle wasting and skeletal muscle strength, given that 28-day sorafenib treatment was applied in our while 42-day treatment was applied by Hout et al. The duration of voluntary wheel running may also partly explain why there were no differences in grip strength and wire-hanging performance between vehicle-treated runners and non-runners. Previous studies have illustrated that seven- to eight-week voluntary wheel running exercise [80,81] can improve grip strength in drug naive wild-type mice, whereas voluntary running regimen lasted for four weeks only in our study. Hence, the duration of chemotherapy administration and voluntary wheel running exercise exposure may contribute to these discrepancies.